

# Field survey of the 2017 Typhoon Hato and a comparison with storm surge modeling in Macau

**Linlin Li[1*], Jie Yang[2,3*], Chuan-Yao Lin[4], Constance Ting Chua[5], Yu Wang[1,6], Kuifeng Zhao[2], Yun-Ta Wu[2], Philip Li-Fan Liu[2,7,8], Adam D. Switzer[1,5], Kai Meng Mok[9], Peitao Wang[10], Dongju Peng[1]**

[1]Earth Observatory of Singapore, Nanyang Technological University, Singapore

[2]Department of Civil and Environmental Engineering, National University of Singapore, Singapore

[3]College of Harbor, Coastal and Offshore Engineering, Hohai University, China

[4]Research Center for Environmental Changes, Academia Sinica, Taipei 115, Taiwan

[5]Asian School of the Environment, Nanyang Technological University, Singapore

[6]Department of Geosciences, National Taiwan University, Taipei, Taiwan

[7]School of Civil and Environmental Engineering, Cornell University, USA

[8]Institute of Hydrological and Ocean Research, National Central University, Taiwan

[9]Department of Civil and Environmental Engineering, University of Macau, Macau, China

[10]National Marine Environmental Forecasting Center, Beijing, China

Corresponding to: Linlin Li (llli@ntu.edu.sg) ; Jie Yang (jie_yang@hhu.edu.cn)

**Abstract:** On August 23, 2017 a Category 3 Typhoon Hato struck Southern China. Among the hardest hit cities, Macau experienced the worst flooding since 1925. In this paper, we present a high-resolution survey map recording inundation depths and distances at 278 sites in Macau. We show that one half of the Macau Peninsula was inundated with the extent largely confined by the hilly topography. The Inner Harbor area suffered the most with the maximum inundation depth of 3.1m at the coast. Using a combination of numerical models, we simulate and reproduce this typhoon and storm surge event. We further investigate the effects of tidal level and sea level rise on coastal inundations in Macau during the landfall of a 'Hato like' event.

## 1 Introduction

On August 23, 2017, at approximately 12:50 pm local time Typhoon Hato made landfall near Zhuhai, which is located on the Southern coast of Guangdong province, China (Figure 1). With an estimated 1-minute sustained wind speed of 185 km/h near its center and a minimum central pressure of 945 hPa, Typhoon Hato was a Category 3



Hurricane on the Saffir-Simpson scale. Typhoon Hato was one of the strongest typhoons to affect the coastal areas of the Pearl River Estuary (PRE) in Southern China over the last several decades. It caused widespread coastal flooding in the PRE areas (ESCAP/WMO Typhoon Committee, 2017). Major cities in the northeast quadrant of the typhoon track, including Macau, Zhuhai and Hong Kong, were severely affected. The resulting maximum storm surge heights (water level above the astronomical tide) reached 1.62 m at A-Ma station in Macau, the highest since water level recording began in 1925. Elsewhere in the PRE areas, a maximum storm surge of 2.79 m was recorded at Zhuhai, and 1.18m, 1.65m and 2.42 m at Quarry Bay, Tai Po Kau and Tsim Bei Tsui in Hong Kong, respectively (HKO, 2017) (Figure 1b). The extreme flooding in Macau was historically unprecedented in terms of the inundation depth as well as the extent, and more than half of the Macau Peninsula was inundated. Typhoon Hato's strong wind and the associated flooding resulted in 22 fatalities and caused 3.5 billion USD direct economic losses (Benfield, 2018).

Macau (and Hong Kong) commonly experience about 5-6 typhoons per year and as the result the low-lying area in the western part of Macau Peninsula has been frequently flooded by storm surges during major typhoons.. Relatively recent typhoons such as Becky (1993), Hagupit (2008), Koppu (2009), and Vicente (2012) all generated storm surges that produced maximum inundation depths > 1 m in Macau, while the unnamed historical typhoons in 1927 and 1948, and typhoon Gloria (1957) generated storm surges > 1.15 m (see the historical flood records http://www.smg.gov.mo/smg/database/e_stormsurge_historicalRec.htm). Although frequently affected by storm surges, the extreme inundation brought by Typhoon Hato still caught Macau unprepared. Consequently, the local government has declared Typhoon Hato as the "worst-case" scenario and will use it as a criterion for designing new engineering measures for coastal protection.

While Typhoon Hato has caused the worst flood event in Macau's history, the key flood parameters (e.g. the water depth and inundation distance) have not been properly documented. Although, Macau has 2 tidtidal levele gauge stations and 17 inland water gauge stations distributed in the areas susceptible to flooding (http://www.smg.gov.mo/smg/ftgms/e_ftgms.htm ). Unfortunately, they all failed to record the peak water level due to breakdown or electricity interruption of devices during Hato (SMG, 2017). Therefore, post event surveys of key flood parameters become essential for better understanding storm surge dynamics and inundation characteristics (e.g. Fritz et al., 2007;Tajima et al., 2014;Takagi et al., 2017;Soria et al., 2016). For this reason, our field survey team was deployed to Macau and Zhuhai on August 26, 2017 and collected flood and damage information for 5 days. Here, the survey data have been analyzed and used to produce a high-resolution inundation map of Macau, which will be discussed in this paper.

Qualitatively speaking, several factors have contributed to the exceptional damage during Typhoon Hato: 1) Typhoon Hato occurred during the second day of a Lunar month and the landfall time roughly coincided with the astronomical high tide; 2) According to the record, Typhoon Hato's wind speed was the strongest among all the typhoons in Macau since 1953. The peak wind gust reached 217.4 km/h in Taipa Grande station and broke the record of 211.0 km/h set by Typhoon Ruby in 1964 (SMG, 2017;Shan et al., 2018). 3) The translation wind speed of Typhoon Hato exceeded 30 km/h (Takagi et al., 2018) before its landfall, which is unusually high compared with the


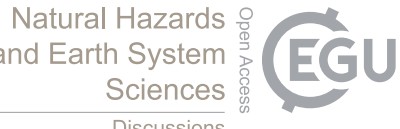

average transitional speed of 10-15 km/h in the South China Sea (Shan et al., 2018). 4) According to Hong Kong
Observatory, Typhoon Hato momentarily became a super typhoon during its approach towards the Pearl River
Estuary in the morning of August 23 (HKO, 2017). The sudden intensification occurred because of the low vertical
wind shear and the high sea surface temperature of ~31 ℃ in the Northern portion of the SCS.
It is well-known that the during a typhoon's landfall plays a significant role in the severity of the storm surge
induced inundation. In the case of Typhoon Hato, the coincidence of astronomical high tide and the landfall time is
thought to be the major factor causing the wide-spread flooding in Macau. However, using the OSU TPXO-atlas8
tide model and the tide gauge location (113.551 °E 22.167 °N) in Macau, we estimated the peak tidal level on the
day of Hato's landfall occurred at 10:00 AM on August 23, 2017 with the tide level of 0.927 m above the mean sea
level (MSL), while the estimated tidal level was only 0.470 m above the MSL at the reported Typhoon Hato's
landfall time around 12:50 pm on August 23, 2017. Thus, Typhoon Hato actually made landfall almost 3 hours after
the peak tidal level, while the tidal level differences are almost 0.5 meter. Thus, it is intriguing to ask what if
Typhoon Hato had occurred at a different time with a lower or higher tidal level, how would the inundation areas
change?
To provide a quantitative answer for the question posted above, a numerical simulation tool must be validated first.
In this paper, the tide-surge-wave coupled hydrodynamic model, SCHISM (Semi-implicit Cross-scale Hydroscience
Integrated System Model) is combined with the Weather Research and Forecasting (WRF) model to simulate the
entire Typhoon Hato event. The high-resolution bathymetric data in PRE and topographic data in Macau are
employed for calculating coastal flooding. Model-data comparisons are performed to ensure that the wind fields are
reproduced well by the WRF model. The field survey data (e.g. inundation depth and area) are used to check the
accuracy of the storm surge model. Once the numerical model is validated, we can use it to conduct a series of
numerical experiments to assess the possible impact of 'Hato-like' typhoon occurring at different tidal levels. Then
looking at such hazard event and its counter-measures from a long-term perspective, we examined the effect of sea
level rise (SLR) on the inundation areas.
The paper is presented in the following order. We first report a high-resolution inundation map of Macau based on
our field measurements and observations. Then we describe each component of the numerical simulation package
followed by the simulation results of Typhoon Hato. Finally, we discuss the effect of tidal level and SLR through the
results of numerical experiments.
**2 Post-typhoon field survey**
On August 26, 2017, three days after Typhoon Hato made landfall, our survey team arrived at Macau, where they
surveyed ~ 300 sites in Macau Peninsula, measuring flow depths (water depth above the street level), maximum
runup, and inundation distances (Table S1). The team also recorded building damage. The team was able to conduct
interviews with many shopkeepers, homeowners and security officers, who witnessed this flood event. The
maximum inundation depths were mainly determined by using watermarks as indicators and where possible





confirmed by eyewitnesses. Watermarks identified on glass panels, iron gates and light colored walls were photographed (Figure 2) and located using GPS. Inundation extent was determined by tracing watermarks from the coastline to the inundation limit along streets perpendicular to the coastline. Distances between two surveyed sites were about 20 - 25 m apart to ensure the high resolution of this survey map. In total, 278 inundation depths were recorded and eyewitnesses confirmed 96 (35%) of them (Table S1).

Figure 3a shows the surveyed inundation depths on the Macau Peninsula. Names of the locations are marked on a high-resolution bare ground elevation map in Figure 3b. The Inner Harbor area, which starts from the A-Ma Temple in the southwest and ends at Qingzhou in the northwest of Macau Peninsula, was completely flooded to a depth of 3.1 m at Ponte Pou Heng on Avenida de Demétrio Cinatti. Along the coastal roads of Rua Visconde Paco de Arcos, Rua do Almirante Sergio and the nearby areas, inundation depth reached 2.0 - 2.5m in many low-lying places. As the seawater penetrated inland, the inundation depth gradually decreased from > 2 m to ~1 m. The inundation extent was clearly confined by the hilly topography (Figure 3a-b). From south to north, the steep topography of the local hills acted as natural barriers, limiting flood propagation. In contrast, the relatively flat (2-3 m above mean sea level) northwest area surrounding "Fai Chi Kei", experienced inundation distances of up to ~1.3 km inland (Figure 3a-b). The coastal area in the northeast was largely spared due to the seawall protection and slightly higher elevation, while the southeast coastal area was slightly flooded by less than 1 m surge with limited inundation distance (<50 m). Considering the size of the Macau Peninsula, which is ~3 km E-W and ~4 km N-S, nearly half of the peninsula was inundated during Typhoon Hato (Figure 3a).

Notably, many eyewitnesses commented that this flooding event was characterized by rapid-rising speed; shopkeepers in the Inner Harbor area stated that seawater rose quickly from the ankle level to chest high in less than 20 minutes, leaving them no time to rescue property or possessions on the ground floor. The ground-floors of most of buildings in Macau are used for commercial purpose, which partly explains why Macau had suffered from economic loss exceeding 1.42 billion USD (HKO, 2017). Although residents who live in the Inner Harbor area are experienced in battling chronic flooding caused by storm surges, the extreme flood caused by Typhoon Hato still came as a surprise for them in many ways (e.g. its speed, depth and extent). In one of the interviews, an elderly resident who lives on the street of Rua Do Camboa used the length of his body as a yardstick to describe the height of floodwater from previous events. He explained that Typhoon Becky (1993) had resulted in approximately 1.4 m floodwater at where he lives; 1.2 m during Typhoon Hagupit (2008); 0.3 m during Typhoon Vicente (2012); and this time Typhoon Hato had resulted in a 2.1 m flood height.

The survey data presented in this study is complementary to the data provided by an earlier study (Takagi et al., 2018) in terms of the number of surveyed locations and spatial coverage. Takagi et al. (2018) provided 12 data points in Macau and Hong Kong while our 278 data points are concentrated in Macau with the purpose of constructing a measured high-resolution inundation map. Such map provides not only valuable documentation of such rare and extreme event but also validation data for numerical modelers.

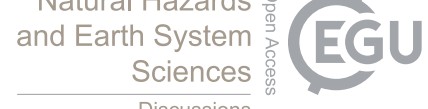

## 3 Numerical simulation


The WRF model (version 3.8.1) (Skamarock et al., 2008) is used to generate the wind and pressure fields of
Typhoon Hato. The initial and meteorological boundary conditions for WRF were obtained from the National
Center for Environmental Prediction (NCEP) Global Forecast System (GFS) with $0.5° × 0.5°$ analysis data sets at 6-
h interval. The time-varying sea surface temperature (SST) was obtained from 0.5° NCEP real-time-global data set.
The planetary boundary layer of the Yonsei University boundary layer scheme (Hong et al., 2006) was used with the
revised MM5 similarity surface layer scheme (Jiménez et al., 2012) and the unified Noah land surface model
(Tewari et al., 2004) over land. The single-moment five-class microphysics scheme (Hong et al., 2004), the updated
Kain-Fritsch scheme (Kain, 2004) with a moisture-advection based trigger function (Ma and Tan, 2009), the Rapid
Radiation Transfer Model for Global Circulation Models (RRTMG) shortwave and longwave schemes (Iacono et
al., 2008) are also adopted in this study. The horizontal resolution of the model is 3 km and the grid box had 921 ×
593 points in both east-west and north-south directions.
The output wind and pressure fields from the WRF results are then used to drive the storm surge simulation in the
tide-surge-wave coupled hydrodynamic model SCHISM (Zhang et al., 2016), which is a derivative code of the
original SELFE (Semi-implicit Eulerian-Lagrangian Finite Element) model. The SCHISM system has been
extensively tested against Standard Ocean/coastal benchmarks and applied to several regional estuaries for storm-
surge inundation modeling (Krien et al., 2017;Wang et al., 2014). In this study, the model is used in the 2DH
barotropic mode, which solves nonlinear shallow water equations on unstructured meshes for storm surges. To track
the coastline movements, the model includes an efficient wetting-drying algorithm by using semi-implicit time
stepping and Eulerian-Lagrangian method for advection (Zhang and Baptista, 2008).
The bottom shear stress is modeled by the Manning's formula with the Manning coefficient being set to 0.025 for
the inland area (area above mean sea level) and 0.01 for the offshore area. The drag coefficient for the surface wind
stress is computed according to Pond and Pickard (1998). The model is forced by applying tidal elevation series on
nodes along open ocean boundaries, which are extracted from the OSU TOPEX/Poseidon Global Inverse Solution
model TPXO8-atlas (Egbert and Erofeeva, 2002).
To capture the effects of wind waves in the storm surge simulation, the spectral Wind Wave Model (WWMIII) is
employed. WWMIII solves the wave action equations in the frequency domain on the same unstructured grid as
SCHISM. Physical processes including wave growth and energy dissipation due to whitecapping, nonlinear
interaction in deep and shallow waters, and wave breaking are all considered in the simulations. The radiation
stresses of the wind wave field are then calculated and used in the storm surge model, SCHISM.
For Typhoon Hato, the simulation domain covers the northern part of the South China Sea (Figure 4a). We create an
unstructured grid with horizontal resolution varying from 50 km in the deep sea, ~ 1 km over the shelf to ~ 20 m in
the vicinity of Macau (Figure 4b). To ensure the accuracy and reliability of the simulation results, we integrated as
many available topographic and bathymetric data as possible: 1) The bathymetric data in the Pearl River Estuary are
integrated from 36 nautical charts with scales ranging from 1:5000 to 1:250000; 2) high resolution topographic data



for Macau is purchased from the Macau Cartography and Cadastre Bureau; 3) 1-arc Shuttle Radar Topography
Mission (SRTM) data covering the Pearl River Delta; (c) 5m elevation is specified for the two artificial islands in
the eastern side of Macau Peninsula, which are still under construction. The topographic and bathymetric data were
complemented by 30 arc seconds General Bathymetric Chart of the Oceans (GEBCO) data and integrated into one
dataset after being adjusted to the mean sea level (MSL).  Using this model setting, we validated the tidal current
model performs well when comparing the simulated tidal cycle with measured data from 1 Nov 2014 to 30 Nov
2014  (Figure S1).
**4 Results**
**4.1 Simulation results of Typhoon Hato**
We first compare the wind speeds generated by WRF with the measured data at 9 selected wind gauge stations in the
PRE (Figure 5c-k), including 4 local wind gauges in Macau (see the gauge locations in Figure 5a-b). The model/data
comparison shows that the WRF model captures Typhoon Hato's wind fields well in terms of both the peak wind
speed and the phase (Figure 5c-k).
The simulated maximum surge heights in the PRE (Figure 6a) show that the storm surge heights on both sides of the
PRE varied widely, ranging from 0 m to 4.5 m. Surge heights > 2.5 m occurred on much of the western side of PRE
including Macau. Wave amplification effects, the funnel-shaped coastline in the PRE also likely led to larger surge
heights in the inner estuary area. To validate the numerical results, we compare the simulated storm-tides with the
measured storm-tides at 4 selected locations (Figure 6b). Very reasonable agreements are observed, ensuring the
reliability of the modelling approach used in this study.
We further compare the simulated and measured inundation maps in Macau (Figure 6c). The calculated inundation
depths are slightly lower (~10%) than the measured ones in some locations near the coastline of "Fai Chi Kei" and
southern Inner Harbor and several inland locations in Inner Harbor. The underestimation is likely the by-product of
the bare-ground topographic data used in the simulation, which does not include buildings, and hence excludes
complex flow patterns (e.g. wave front colliding with buildings) and channeling effects, which locally increase
water depth. Nevertheless, the overall agreement is quite good. It demonstrates that the coupled model can
reproduce this flood event reasonably well in terms of both inundation depth and extent.
In Figure 6d we also plot the simulated arrival time of wave front, which can be viewed as the arrival time of the
surge.  During Typhoon Hato the surge wave arrived in the southern Inner Harbor first at the local time around
11:20 on August 23, 2017 and then propagated eastward inland and northward in the Fai Chi Kei direction in the
next one and half hours. The flow velocity was less than 0.5 m/s in the southern Inner Harbor due to the generally
steep slope, while in the area surrounding Fai Chi Kei, the flow velocity was faster and was up to 0.7 - 0.8 m/s.
When comparing with the 2 – 5 m/s tsunami flow velocity recorded in the Banda Aceh during the 2004 Indian
Ocean tsunami (Fritz et al., 2006), the 0.5-0.8 m/s flow velocity of storm tide in Macau is significantly smaller. The





information like this demonstrates that numerical model and field survey can complement each other and recover a comprehensive view of disaster scene.

**4.2 The effects of tidal level**

Having checked our numerical model with measured data and demonstrated that the model can replicate events like Typhoon Hato, we now investigate the effects of tidal level on coastal flooding. Most the PRE including Macau has a mixed semidiurnal tidal cycle in which the semidiurnal lunar tide, M2, is the predominant component followed by K1, O1 and S2. The maximum tidal range observed in Macau is 2.86 m while the difference between mean high water (MHW) and mean low water (MLW) is 1.11 m (calculated from the tide record during 1985-2012).

To quantitatively investigate the impact of tidal level at Hato's landfall, we first selected two extreme tidal levels from the years of 1964-2017 using OSU TPXO-atlas8 tide model. The reason we use more extreme tidal levels is because scenarios under those tidal levels can provide the upper and lower bounds of the potential inundations, thus better demonstrating the effect of tidal level. On the other hand, we observe the peak tidal level on the day of Typhoon Hato's landfall was only moderately high compared with the daily higher high water records (HHW) in Macau (Figure S2) although it was the third highest during that month. Putting this peak tide of 0.927 m in all the estimated daily HHW, we can see that this value is lower than 21% of the daily HHW during 1964-2017 (Figure S2 shows the HHW and LLW during 2008-2017 as an example). We find the corresponding highest extreme tide (HET) and the lowest extreme tide (LET) occurred on January $1^{st}$, 1987 at 22:00 and January $2^{nd}$, 1987 at 6:00, respectively with the tide 1.304 m above MSL and 1.165 m below MSL. We then conduct storm surge simulations at these two selected extreme tidal levels by moving the typhoon landfall time from 13:00 (UTC+8), Aug 23, 2017 to January $1^{st}$, 1987 at 22:00 and January $2^{nd}$, 1987 at 6:00 local time.

Figure 7 shows the maximum inundation maps for the real case (benchmark scenario, Figure 7a), at HET (Figure 7b) and LET (Figure 7c), respectively. The striking observation is that Typhoon Hato would cause inundation in Macau at all the considered tidal levels (Figure 7a-c). Even Typhoon Hato occurred during the LET, the Inner Harbor area would still be inundated with the maximum inundation depth up to 1.0 m (Figure 7c). We emphasize here, as the LET is a representative value of the lowest extreme tidal level in the past 54 years (1964-2017), which suggests the Inner Harbor area will certainly be inundated during Hato-like events regardless of the tidal levels at the landfall time. The results once again highlighted the vulnerability of the Inner Harbor area to extreme flooding and urgency of establishing effective protection system. Not surprisingly, if Hato had occurred at HET, a noticeably greater inundation depth (0.5-0.8 m deeper in the Inner Harbor, see Figure 8) would have been sustained in all the flooded areas on the Macau Peninsula with inundation extending considerably to previously unaffected areas. For example, had Hato struck at the HET, the northeast Macau Peninsula, the coastal area of Taipa and Cotai would be inundated with up to 1-m water depth or higher.



### 4.3 Investigation sea level rise

To account for the effects of future sea level rise (SLR), the values of 0.5 m and 1 m SLR were chosen to represent the sea levels by the mid-century and end of this century based on projected local sea level rises of 30-51 cm by 2060 and 65-118 cm by 2100 (Wang et al., 2016). We then ran the storm surge simulations at HET and LET with different magnitude of SLR 0.5 m and 1.0 m. For each simulation, we obtain the maximum inundation depths at all in-land computational nodes by subtracting the DEM data from the simulated maximum wave heights. In total, we derive three sets of inundation maps at current sea level, 0.5 m and 1.0 m SLR condition.

Adding the effect of 0.5-m and 1-m SLR at HET, the inundation extent quickly expands into the eastern part of the Macau Peninsula and coastal areas of Taipa, Cotai and University of Macau, where no or limited flood was observed during Typhoon Hato (Figure 9a-b). Compared with the benchmark scenario (Figure 7a), the maximum inundation depths in inner harbour area will increase more than 1-1.2 m in the 0.5-m SLR scenario (Figure 9e) and 1.2-1.5 m in the 1-m SLR scenario for most of the inner harbour area (Figure 9i); such increase is generally a linear combination of increased tidal level and SLR. While in the eastern side of Macau Peninsula and some places in Taipa and Cotai, we observe larger increases in the water depths than in the inner harbour area. The increased water depths can be up to 1.2-1.5 m and 1.5-2.0 m at 0.5-m and 1-m SLR conditions, respectively. The larger increase can be partly attributed to large waves in the eastern side of Macau Peninsula, coastal areas of Taipa and Cotai than in the inner harbour area, especially at higher sea level conditions (Figure 10). Such spatially non-uniform response of storm waves to SLR has been discussed in previous studies in many coastal areas worldwide (e.g. Atkinson et al., 2013;Bilskie et al., 2016;Mcinnes et al., 2003) and China (e.g. Wang et al., 2012;Yin et al., 2017). Comparing the maximum inundation depth between scenarios at LET under different sea level conditions (Figure 9c-d), we point out once again that the Inner Harbor area will suffer increasingly more hazardous inundation with the rising sea level. Thus engineer measures are urgently required to protect this area. When designing such engineer measures, proactive policies and adaptive strategies should be taken to combat the likelihood of worsening flooding in future.

### 5 Conclusions

Typhoon Hato was one of the most damaging natural disaster events in the Western Pacific region in 2017. It caused extensive coastal inundation in and around the Pearl River Delta region. In this paper, we have presented a detailed post-typhoon field survey, yielding 278 measurements of maximum water depths and inundation extent on Macau Peninsula. Using these data, a high-resolution flood map has been produced. These survey data have been used to successfully validate a numerical model package, which consists of a WRF model for calculating the wind and atmospheric pressure field and a tide-surge-wave coupled hydrodynamic model, SCHISM, for computing tides, storm surges and ocean waves driven by the WRF model results. The data-model comparisons show that the skills of the numerical model package are high and can capture all key features of this event, including the wind fields, the water levels associated with storm surges and tides in the PRE, and the inundation depths in Macau. More



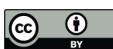

importantly, the numerical model package can provide additional information such as the arrival times of the storm
surge front and the corresponding shoreline movements.
The numerical model package can also be used to gain better understanding of the relative importance of different
causes for coastal flooding. To demonstrate this capability, we have focused on studying the effects of tidal level
and SLR on coastal inundation in Macau, using Typhoon Hato's atmospheric condition as a benchmark scenario..
One of the important observations is that regardless the tidal level during Typhoon Hato's landfall, the Inner Harbor
area will always be inundated with the maximum inundation depth up to 0.5 -1.0 m. On the other hand, although
Typhoon Hato broke all the historical records in terms of storm surge heights and flooded area, much worse scenario
could be expected if Typhoon Hato had occurred at a higher tidal level, and thus, caution is required if Typhoon
Hato is to be used as the worst-case scenario for designing future coastal defense measures. This is especially true
when taking the rising sea level into consideration as 0.5-m and 1-m SLR could significantly increase the severity of
the resulting inundation for most of the territory in Macau, including both the high tide and low tide conditions. The
inundation maps presented in this study provide the lower and upper bound of potential impacts of Hato-like events
at different tidal levels and sea level conditions. Such maps could aid the local government to make more
informative decisions.
Besides the tidal level, other factors including the landing location, track azimuth, forward speed, the sudden
intensification and urban development (e.g. land reclamation) may play more important role in contributing to this
record-breaking flood in Macau. The effects of such factors are being analyzed in more detail in a future paper.
Hato-like typhoon events pose a clear and significant threat to the emerging mega-cities area of the PRD and the
drive to expand towards the sea with extensive land reclamation and infrastructure development needed to meet the
demands of the growing population and the booming economy. Although most major cities in the region are
protected by seawalls, the protection standards vary considerably and whether such standards are sufficient to
combat increasingly frequent flooding in future needs careful investigation. Adaptive strategies and sustainable
management are almost certainly required in order to keep up with the pace of rising sea level. We believe that the
data and findings provided in this paper and the numerical model package will not only be of great interest to coastal
hazard researchers, but also to a range of stakeholders such as policy makers, town planners, emergency services
and insurance companies who are working to create or insure safer coastlines.

**Acknowledgments, Samples, and Data**

We are very grateful to the Macau people for the extremely helpful information, photographs, video footages
provided, and the kindness they have shown us during the field survey. We thank Dr. Hoi Ka In for the tide data
analysis and Dr Zhiguo Ma for helping processing the topographic data of Macau. We thank the Macau
Meteorological and Geophysical Bureau for providing us with the meteorological data of Macau. This study is
supported by AXA Research Fund Post-Doctoral Fellowship under the project "Probabilistic assessment of multiple
coastal flooding hazards in the South China Sea under changing climate" to Linlin Li and the Ng Teng Fong
Charitable Foundation (Hong Kong) under the joint research project "The impact of climate changes on coastal



flooding hazard in South and East China Seas" between National University of Singapore and Tsinghua University.
Adam Switzer was supported by (AcRF) Complexity Tier 1 Project RGC4/14 "Preparing Asian mega cities for
changing climate and the potential Increase in extreme sea levels and storm surges". This paper contributes to
IGCP639 "Sea level change: from Minutes to Millennia". The navigational charts in the Pearl River Estuary are
purchased from Beijing Situo Ocean Information Technology Co Ltd. The topographic data of Macau was
purchased from the Macau Cartography and Cadastre Bureau. The GEBCO data used in this study is downloaded
from http://www.gebco.net in October 2014.

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

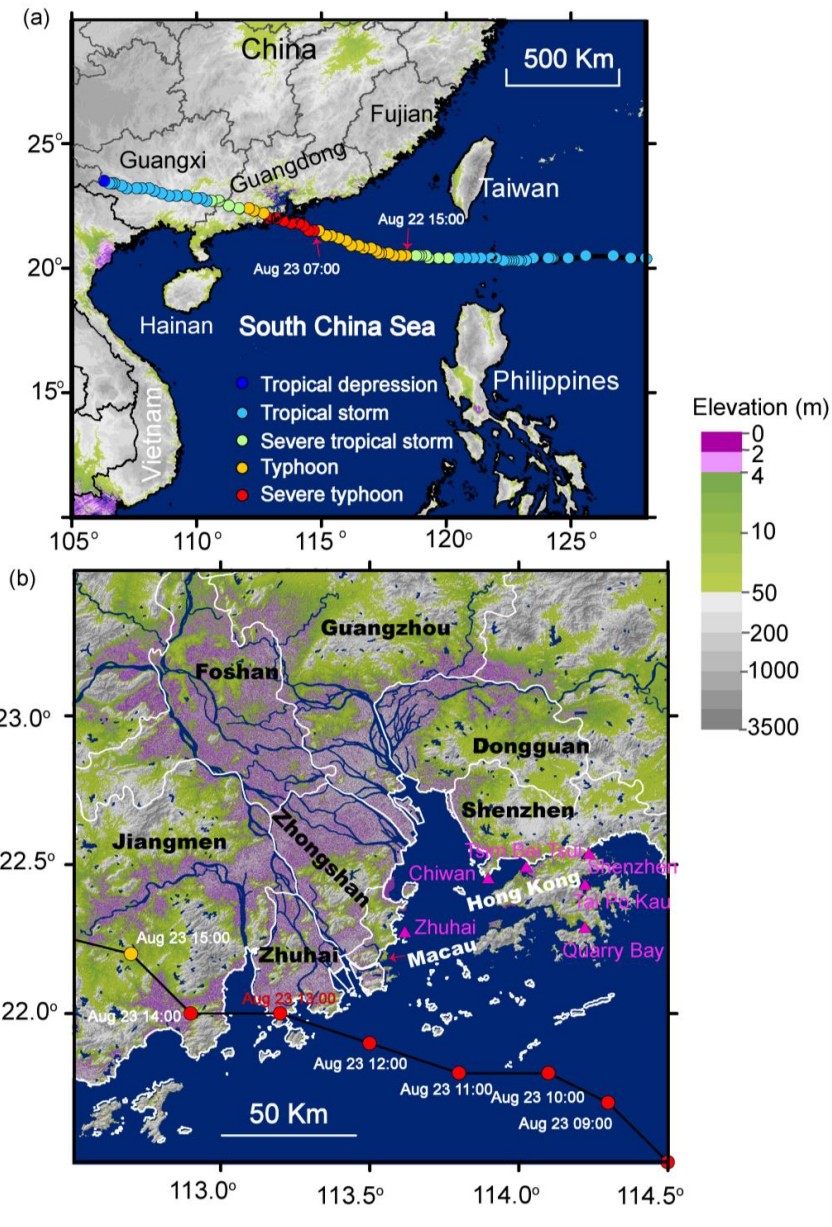


**Figure 1.** Typhoon Hato track data from the Chinese typhoon weather website (http://typhoon.weather.com.cn/ ). (a)

Typhoon Hato took a path extremely dangerous to the Pearl River Delta. It became a typhoon inside the South China

Sea at 15:00 on August 22, 2017 and further was intensified into a Severe Typhoon at 07:00 AM on August 23

before making landfall at 12:50 PM in southern part of Zhuhai, China. (b) A close-up shows the landfall location




392  and the affected cities in the Pearl River Delta. Purple colors denote land elevation lower than 4 m above mean sea

393  level.

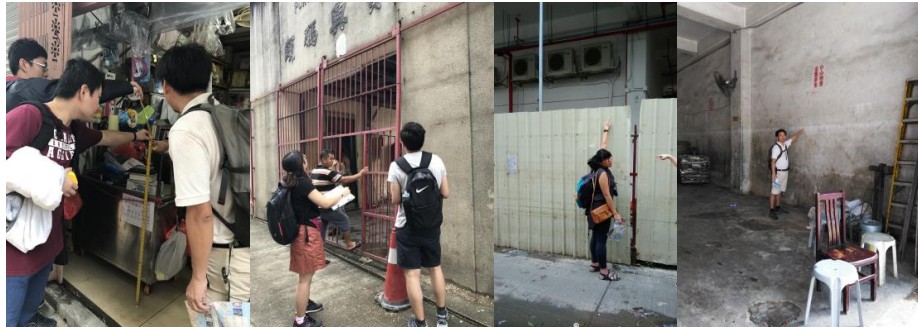

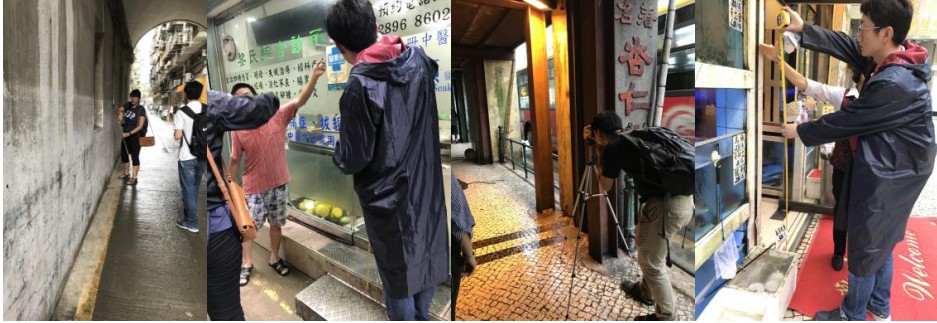

396  **Figure 2.** Photos taken during the field survey on the Macau Peninsula.




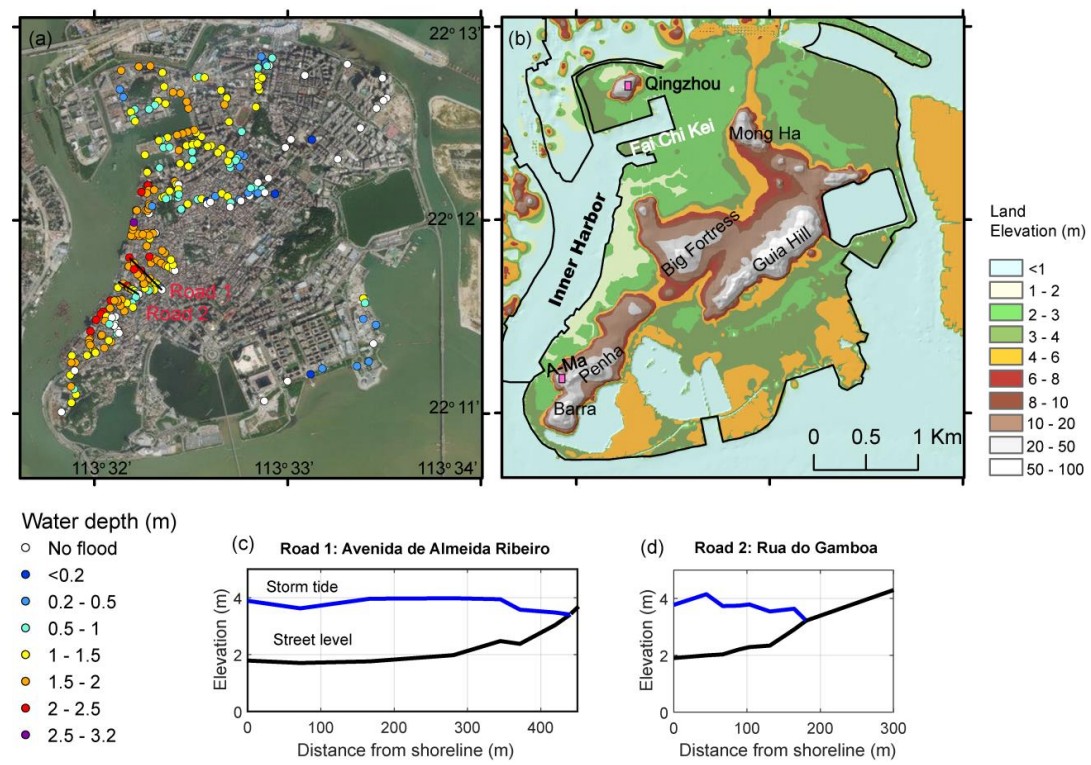

**Figure 3.** (a) Measured inundation depths on the Macau Peninsula shown on a Google Earth image. (b) High-resolution bare ground elevation with marked locations. (c) and (d) Profiles of surveyed inundation water depths along two main roads: Avenida de Almeida Ribeiro and Rua do Gamboa.

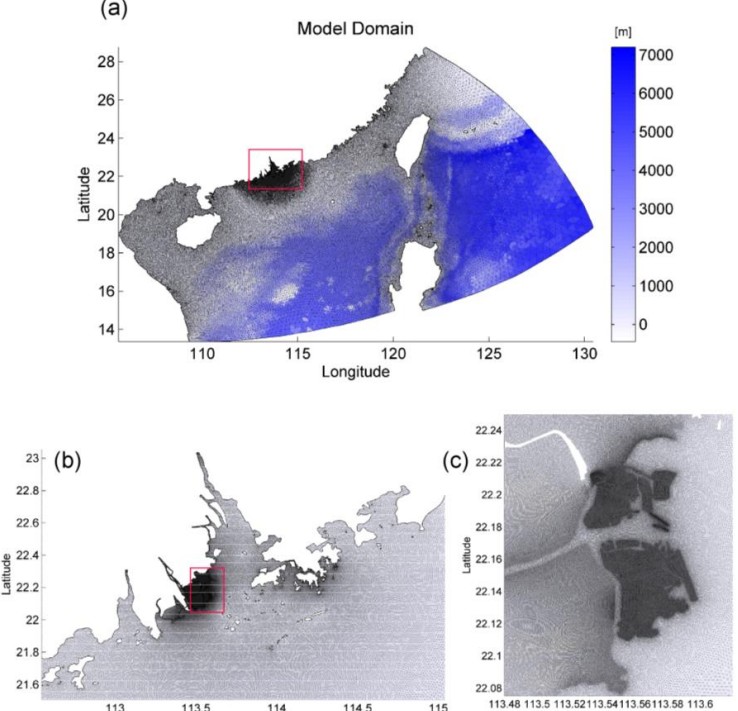

**Figure 4.** (a) The numerical simulation domain for SCHISM-WWMIII with close-ups showing (b) the mesh in PRD
and (c) the mesh near Macau.





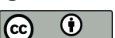
**Figure 5.** Wind fields generated by the WRF model: (a) In the Pearl River Estuary at 12:50 PM on August 23; (b)
The wind gauge locations of PG, PN and PV in Macau; (c) – (k) Comparisons of numerical results (WRF) with
measured wind speed at different locations. Locations and names of wind gauges (c) - (h) are shown in Figure 5a.

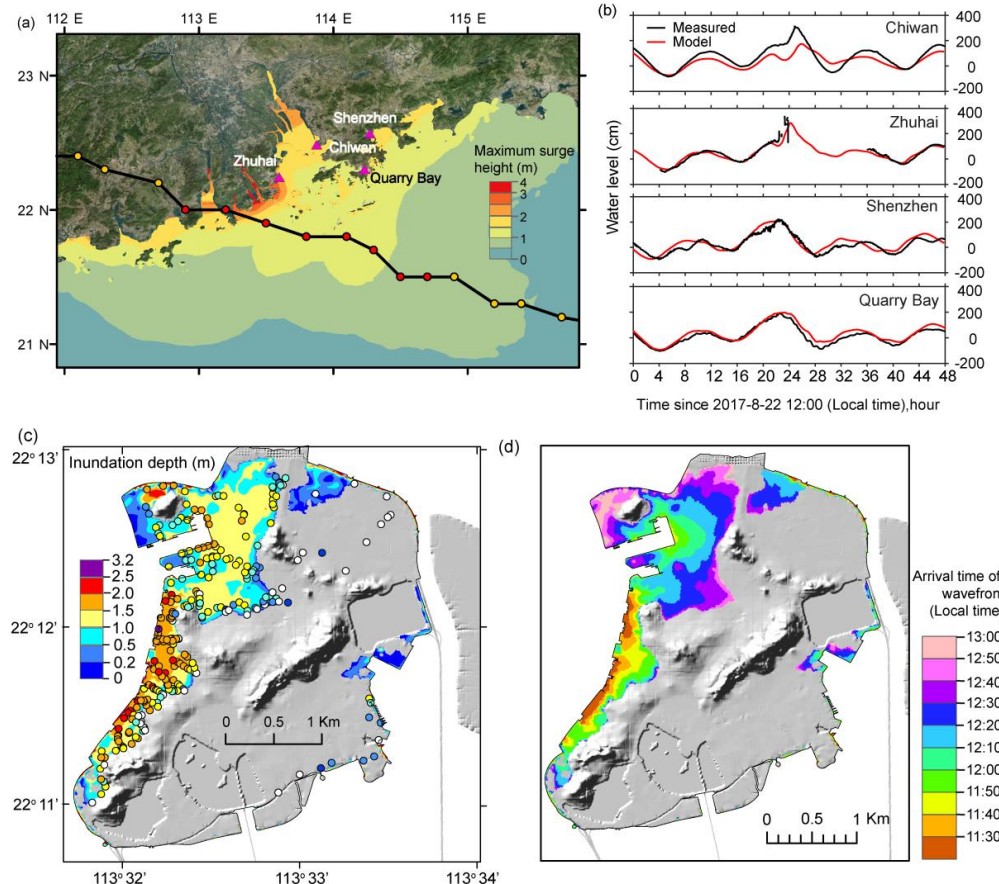


**Figure 6.** Numerical results capture well the key features of storm flooding induced by Typhoon Hato. (a) The
simulated maximum surge height in the PRE; (b) A comparison of simulated and measured storm tide at four
selected tide locations (marked with the green dots in Figure 5a). Note the measured data at Zhuhai station is not
complete due to a power cut; (c) The surveyed inundation depths (the colored dots) overlaid on the simulated
maximum inundation depths in the Macau Peninsula; (d) The arrival time of the flood wavefront.




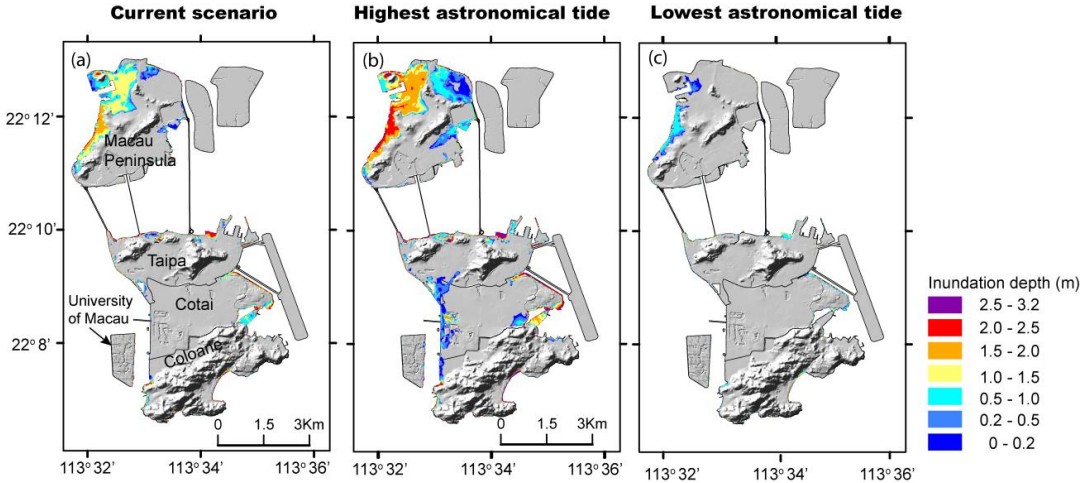


**Figure 7.** Maximum inundation depths for (a) the benchmark scenario; (b) the highest extreme tide and (c) the
lowest extreme tide under the current sea level.

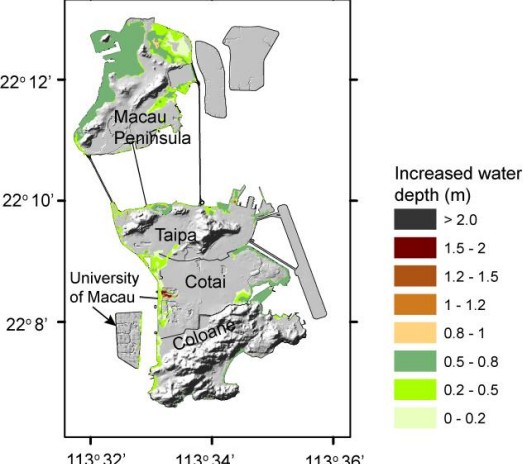


**Figure 8.** A map showing the difference between the maximum inundation during the benchmark scenario (Figure
5a) and the highest extreme tide under the current sea level.



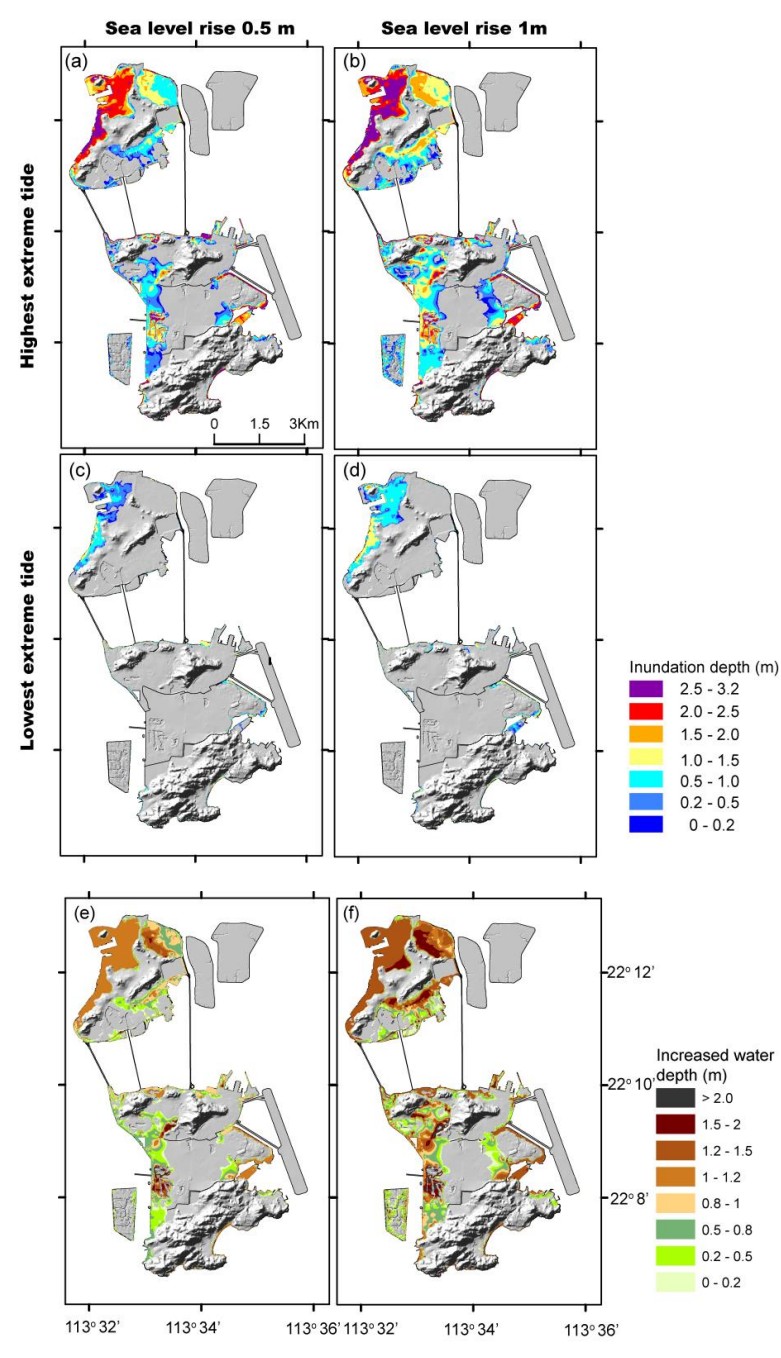



**Figure 9.** Maximum inundation depths during the highest extreme tide under (a) 0.5-m SLR and (b) 1-m SLR.
Maximum inundation depth during the lowest extreme tide under (c) 0.5-m SLR and (d) 1-m SLR; A map showing





the difference between the maximum inundation during the benchmark scenario (Figure 7a) and the highest
extreme tide under (e) 0.5-m SLR and (f) 1-m SLR.

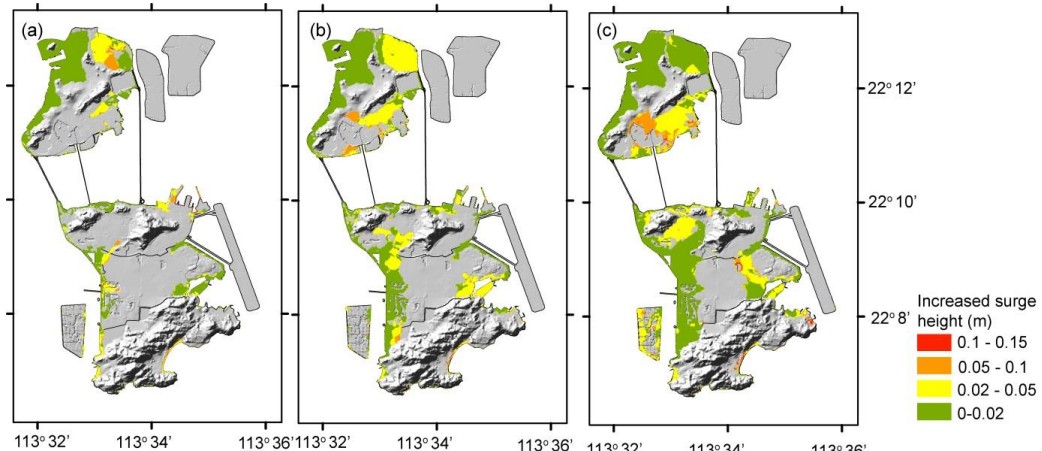


**Figure 10.** The difference in maximum inundation depths between scenarios with and without the wave model
during the highest extreme tide under (a) the current sea level (b) 0.5-m SLR, and (c) 1-m SLR.