# Peer review of "Field survey of the 2017 Typhoon Hato and a comparison with storm surge modeling in Macau"

_Natural Hazards and Earth System Sciences, 2018_

## Referee Comment (RC1) · Anonymous Referee #1 · 26 Sep 2018

In this study, authors conducted immediate post-typhoon field survey in Macau right after Typhoon Hato in 2017, which significantly influenced Macau city. This is rather important for disaster prevention and mitigation efforts in the future since it can provide lots of first-hand information. Afterwards, authors developed a numerical model to simulate the event and validated the model using their own survey data. Then, some further numerical studies were carried out to investigate different scenarios. In general, the MS is well prepared and written. After minor revision, I recommend the immediate publication of the paper considering that another similar typhoon Mangkhut (No 1822) occurred in 2018, which again affected the Macau city. These two cases could be inter-compared to explore many interesting phenomena and physical insights to help the local government to do a better countermeasure against such typhoon disasters.

[Figure]

Detailed comments include, 1. Lines 70-73. This is an interesting point. In general, the maximum storm surge occurs before the typhoon landfall. Hence, the worst scenario is the high tide occurs several hours before the typhoon landing. Ref.: Lai, F., Liu, H. (2017) Wave setup properties in the surge-wave coupled simulation: A case study of Typhoon Khanun. IUTAM symposium on storm surge modelling and forecasting, Procedia IUTAM, 25, 111-118. 2. Lines 104-112. Interesting to see that inundation mainly occurred in the west part of the Macau Peninsula. This, on one hand, is caused by the low-lying topography in the west as mentioned by authors. On the other hand, the southeast region is directly facing the Pacific Ocean and typhoon attack which, in my mind, may suffer more severe wave actions comparing to the west region of the peninsula (though its elevation may be higher than west region). According to Fig 3(a), there is a S-N directed breakwater located in the southeast, which may protect the southeast region to some extent. Could authors specify these more in detail? As for northeast region, it may be protected/shadowed by the islands located in the east. Any descriptions about Figs 3(c) and 3(d)? 3. Lines 124-125. Could authors add Takagi's survey data of Macau in Fig 3a, just for comparison? 4. Fig 4(c) is not mentioned in the context. 5. For section 4.3. Just a suggestion. According to IPCC, the intensity of typhoon will also increase, accompany with the SLR. Hence, authors may apply the scenario under which the typhoon intensity is enhanced together with different SRL. 6. Lines 269-270. This may be not suitable since according to IPCC, the frequency and intensity of future typhoon is increasing. Hence, the worst-case scenario of future typhoon should be more severe than typhoon Hato. 7. In conclusion, please point out clearly that the inner harbor area is the most fragile region which could be inundated even under the lowest tidal level. For this, immediate attentions/engineering actions should be took by the local government.

---

## Referee Comment (RC2) · Anonymous Referee #2 · 20 Oct 2018

The author conducted a field survey and made an inundation map indicating the depth and distance of flooding in the coastal areas of Macau. In addition to the field survey, the author also used the atmosphere model, storm surge model, and wave model (WRF + SCHISM + WWM3) to simulate and reproduce the event as the benchmark scenario and compared with the real event. In this paper, the benchmark scenario is used to investigate the situation which tide and sea level rise affect the events like HATO. In terms of the change in tide level, even if the typhoon HATO is landing in Macau at the time of the lowest tide record, there will still be a flood in the Inner Harbor area, indicating the urgency of adjusting the disaster prevention measures. In terms of sea level rise, the relevant situational simulations show that sea level rise will increase the overflow area. In addition to the linear superposition of water levels, there may be

some reasons that the coastal waves are caused by some non-uniform spatial changes caused by rising water levels. This part is consistent with the literature.

Some minor commands are listed as below: 1. line 17-23: The purpose of this paper and the conclusion are suggested to be included in the abstract. 2. line 39: "Macau (and Hong Kong) commonly experience about 5-6 typhoons per year...", references are required. 3. line 40: "...by storm surges during major typhoons.." The periods are doubled. 4. line 49: "...Although, Macau has 2 tidtidal levele gauge". Typo. 5. line57: "which will be discussed in this paper." Redundant. . 6. line 66-67: "The sudden intensification occurred because of the low vertical 67 wind shear and the high sea surface temperature of ∼31°C in the Northern portion of the SCS". References are required. 7. line 68: "It is well-known that the ... during a typhoon s landfall plays... ". Missing words. 8. line 70: "OSU TPXO-atlas8". References are required. 9. line 79-80...: "SCHISM (Semi-implicit Cross-scale Hydroscience Integrated System Model)..." 10. line 80: "Weather Research and Forecasting (WRF) model..."References are required. 11. Fig 2: Table S: The location of the photo shall beaded in the figure. 12. Fig 3a: The red texts are not clear. 13. line 106-107: The locations, Ponte Pou Heng on Avenida de Demétrio Cinatti, Rua Visconde Paco de Arcos, Rua do Almirante Sergio, are missing on the figures. 14. Fig 3b: No 'Rua Do Camboa' on the firure. 15. line 133-143: The information of timestep, domain, vertical resolution used in WRF shall be explained. 16. line 152-154: The way to determine the Manning coefficient shall be explained with related references. 17. line 160-161: How to setup the radiation stresses in SCHISM? 18. Fig 5a: The text on the figure are not clear enough. 19. Figure 6b : What are the reasons that the error at Chiwan is larger than othe stations?. 20. line 195, Fig 6c, Fig 6d: The location name cannot be found on the figures.

---

## Author Comment (AC2) · 24 Oct 2018

We thank reviewer 2 for the valuable suggestions. In this revised version, we have made changes according to the suggestions and comments and highlighted areas where those changes are made. The point-by-point replies to the comments are below.

Comment 1: line 17-23: The purpose of this paper and the conclusion are suggested to be included in the abstract.

Author's response: Thanks for the suggestion, we actually included the purpose and conclusion in our initial version, but the abstract for NHESS has 100 word limits, so we shortened it to current version.

Author's change to manuscript: no change is made.

[Figure]

Comment 2. line 39: "Macau (and Hong Kong) commonly experience about 5-6 typhoons per year. . .", references are required.

Author's change to manuscript: we add the reference: Lee, E. K. S., L. Fok, and H. F. Lee (2012), An Evaluation of Hong Kong's Tropical Cyclone Warning System, Asian Geographer, 29(2), 131-144.

Comment 3. line 40: ". . .by storm surges during major typhoons.." The periods are doubled.

Author's change to manuscript: Extra "." deleted.

Comment 4. line 49: ". . .Although, Macau has 2 tidtidal levele gauge". Typo.

Author's change to manuscript: Corrected.

Comment 5. line57: "which will be discussed in this paper." Redundant.

Author's change to manuscript: Redundant sentence is deleted.

Comment 6. line 66-67: "The sudden intensification occurred because of the low vertical 67 wind shear and the high sea surface temperature of around 31 Celsius degree in the Northern portion of the SCS". References are required.

Author's change to manuscript: we add the reference: HKO: Super Typhoon Hato (1713), Hong Kong Observatory, http://www.weather.gov.hk/informtc/hato17/report.htm, 2017.

Comment 7. line 68: "It is well-known that the . . . during a typhoon landfall plays ËŸ . . . ". Missing words.

Author's change to manuscript: we added the missing word "tide". Here is the corrected sentence: "It is well-known that the tide during a typhoon's landfall plays a significant role"

Comment 8. line 70: "OSU TPXO-atlas8". References are required.

[Figure]

Author's change to manuscript: we add the reference: Egbert, G. D., and S. Y. Erofeeva (2002), Efficient Inverse Modeling of Barotropic Ocean Tides, Journal of Atmospheric and Oceanic Technology, 19(2), 183-204.

Comment 9. line 79-80. . .: "SCHISM (Semi-implicit Cross-scale Hydroscience Integrated System Model). . ."

Author's change to manuscript: we add the reference: Zhang, Y. J., F. Ye, E. V. Stanev, and S. Grashorn (2016), Seamless cross-scale modeling with SCHISM, Ocean Modelling, 102, 64-81.

Comment 10. line 80: "Weather Research and Forecasting (WRF) model. . ."References are required.

Author's change to manuscript: We add the reference: Skamarock, W. C., Klemp, J. B., Dudhia, J., Gill, D. O., Barker, D., Duda, M. G., Huang, X.-y., Wang, W., and Powers, J. G.: A Description of the Advanced Research WRF Version 3, National Center for Atmospheric Research, Boulder, Colorado, USA, NCAR TECHNICAL NOTE, 2008.

Comment 11. Fig 2: Table S: The location of the photo shall be added in the figure.

Author's response: We added the longitude and latitude of each photo in the figure.

Author's change to manuscript: Please refer to Fig 2 for the changes.

Comment 12. Fig 3a: The red texts are not clear.

Author's change to manuscript: We changed the colour to yellow which we believe is clearer.

Comment 13. line 106-107: The locations, Ponte Pou Heng on Avenida de Demétrio Cinatti, Rua Visconde Paco de Arcos, Rua do Almirante Sergio, are missing on the figures.

Author's response: We realize it might be very difficult for readers who are not familiar

with the roads' names to identify them on the figure. Therefore, instead of adding the names on the map, we chose to remove the names and use other features already on the maps to describe the area referred to.

Author's change to manuscript: The sentence with the listed names is change to "...was completely flooded to a depth of 3.1 m at Ponte Pou Heng (the purple dot). Along the coastal roads of the Inner Harbor, inundation depth reached 2.0 - 2.5m in many low-lying places."

Comment 14. Fig 3b: No 'Rua Do Camboa' on the figure.

Author's response: The street "Rua Do Camboa" is shown in Fig 3a and Fig 3d.

Comment 15. line 133-143: The information of time step, domain, vertical resolution used in WRF shall be explained.

Author's change to manuscript: We added more explanation in the manuscript: "There were 45 vertical levels with the lowest level approximately 50 m above the surface. The output time interval of wind and pressure fields is 10 minutes."

Comment 16. line 152-154: The way to determine the Manning coefficient shall be explained with related references.

Author's change to manuscript: We added the requested information in the manuscript: "The values of Manning coefficient are informed by previous studies (e.g. Martyr et al., 2013; Garzon and Ferreira, 2016). We choose relatively low Manning value for the estuary and open sea area as the sediment in the Pearl River Estuary is dominated by very fine sand (mainly silt clays) (Jiang et al., 2014)."

17. line 160-161: How to setup the radiation stresses in SCHISM?

Author's change to manuscript: The WWMIII is dynamically coupled with SCHISM every 600 seconds. The radiation stress is estimated according to Roland (2008) based on the directional spectra itself. The radiation stresses computed in WWMIII are transferred to SCHISM at each step to update water level and velocity, which are returned to WWWIII as feedback.

18. Fig 5a: The text on the figure are not clear enough.

Author's change to manuscript: We have modified the figure by adjusting the colour of coastline and station name. The texts now stand out from the background colours.

19. Figure 6b : What are the reasons that the error at Chiwan is larger than other stations?

Author's response: The tidal gauge in Chiwan is located deep inside of the bay area (see the figure below) where the bathymetry and coastal geometry were subjected to significant changes in the past decade. The bathymetric data used in the simulation does not necessarily capture the most updated status in the bay.

20. line 195, Fig 6c, Fig 6d: The location name cannot be found on the figures.

Author's response: The two locations "Inner Harbor" and "Fai Chi Kei" are marked on Fig 3b. Readers can refer to Fig 3b for the locations.

Author's change to manuscript: We add the sentence "(see the locations in Figure 3b)" in the main text for readers' convenience.

References:

Garzon, J., and Ferreira, C.: Storm Surge Modeling in Large Estuaries: Sensitivity Analyses to Parameters and Physical Processes in the Chesapeake Bay, Journal of Marine Science and Engineering, 4, 45, 2016. Jiang, S., Xu, F., Li, Y., Liu, X., Zhao, Y., and Xu, W.: Distributional characteristics of grain sizes of surface sediments in the Zhujiang River Estuary, 30-36 pp., 2014. Martyr, R. C., Dietrich, J. C., Westerink, J. J., Kerr, P. C., Dawson, C., Smith, J. M., Pourtaheri, H., Powell, N., Ledden, M. V., Tanaka, S., Roberts, H. J., Westerink, H. J., and Westerink, L. G.: Simulating Hurricane Storm Surge in the Lower Mississippi River under Varying Flow Conditions, Journal

[Figure]

of Hydraulic Engineering, 139, 492-501, doi:10.1061/(ASCE)HY.1943-7900.0000699, 2013. Roland, A.: Development of WWM II: Spectral wave modeling on unstructured meshes, 2008.

**Fig. 1.** The location of Chiwan gauge

---

## Author Response (AR1)

Response to Reviewer 1

We thank reviewer 1 for the valuable suggestions. In this revised version, we have made changes according to the suggestions and comments and highlighted where those changes are made.

The point-by-point replies to the comments are below.

**General comments:** In general, the MS is well prepared and written. After minor revision, I recommend the immediate publication of the paper considering that another similar typhoon Mangkhut (No 1822) occurred in 2018, which again affected the Macau city. These two cases could be inter-compared to explore many interesting phenomena and physical insights to help the local government to do a better countermeasure against such typhoon disasters.

**Author's response:** Thank you for your encouraging comments. Your suggestion of making comparison between Typhoon Hato (2017) and Typhoon Mangkhut (2018) is very important. Both Typhoons were rare and record-breaking events in terms of their extreme wind speeds and wide-spreading coastal flooding they caused in the Pearl River Delta region. Actually, immediately after Typhoon Mangkhut (2018), some of our co-authors did another post-typhoon field survey and obtained the first-hand flood parameters in the same area in Macau. The comparison between these two typhoons and their associated physical phenomena is on-going and will be discussed in a future paper.

**Comment 1**: Lines 70-73. This is an interesting point. In general, the maximum storm surge occurs before the typhoon landfall. Hence, the worst scenario is the high tide occurs several hours before the typhoon landing. Ref.: Lai, F., Liu, H. (2017) Wave setup properties in the surge-wave coupled simulation: A case study of Typhoon Khanun. IUTAM symposium on storm surge modelling and forecasting, Procedia IUTAM, 25, 111-118.

**Author's response:** We respect the point that the maximum storm surge generally occurs before the typhoon landfall. However, we also feel that the timing of maximum surge and typhoon landfall depends on many other factors, for example, location of the study area relative to the typhoon track, typhoon size, etc. In the case of Typhoon Hato, the maximum surge indeed occurs before the typhoon landfall, but it is only ~20 minutes before the landfall and did not exactly coincide with the peak tide on that day. Hence, Typhoon Hato could have caused worse flooding if the landing time was ~10:00 AM. To demonstrate this, we simulated a scenario that shifts the landfall time 3 hours earlier than the real case. It shows that the water depth could be 0.2-0.5 m deeper than the real case.

[Figure]

**Comment 2:** Lines 104-112. Interesting to see that inundation mainly occurred in the west part of the Macau Peninsula. This, on one hand, is caused by the low-lying topography in the west as mentioned by authors. On the other hand, the southeast region is directly facing the Pacific Ocean and typhoon attack which, in my mind, may suffer more severe wave actions comparing to the west region of the peninsula (though its elevation may be higher than west region). According to Fig 3(a), there is a S-N directed breakwater located in the southeast, which may protect the southeast region to some extent. Could authors specify these more in detail? As for northeast region, it may be protected/shadowed by the islands located in the east. Any descriptions about Figs 3(c) and 3(d)?

**Author's response:** Yes, the elevation difference between the west and the east part of the Macau Peninsula is the key reason why these two areas experienced different flood levels. The elevation in the Inner Harbour area is only 1-2 m above the mean sea level (MSL). In contrast, the elevation in the southeast is 4-6 m above MSL, and northeast is 3-4 m above MSL. The N-S directed object shown in Fig 3(a) is one of the three bridges connecting the Macau Peninsula with the island to the south, not a breakwater. So the water can pass through the bridges underneath. We apologize for the confusion and we added the text "Bridge" in Fig 3(a) to avoid the confusion. For the northeast part of the Macau Peninsula, the newly reclaimed islands do play a protective role.

We add more descriptions about Fig 3(c) and 3(d) in the manuscript: When tracing the watermarks along the two major streets: Avenida de Almeida Ribeiro and Ruo Do Gamboa, we observe that, as the seawater penetrated inland, the inundation depth gradually decreased from > 2 m to ~1 m (Fig 3c-d).

**Comment 3:** Lines 124-125. Could authors add Takagi's survey data of Macau in Fig 3a, just for comparison?

**Author's response:** To avoid confusion, we plotted a separate figure for Takagi's survey data instead of adding them to Fig 3a. The figure is plotted with the same colour scale of Fig 3a and added as a supplementary Figure S3.

[Figure]

Figure S3. The inundation depths surveyed by Takagi et al. (2018).

**Comment 4.** Fig 4(c) is not mentioned in the context.

**Author's response:** We added it in the third paragraph of Section 3 Numerical simulation.

**Comment 5.** For section 4.3. Just a suggestion. According to IPCC, the intensity of typhoon will also increase, accompany with the SLR. Hence, authors may apply the scenario under which the typhoon intensity is enhanced together with different SRL.

**Author's response:** Thanks for the suggestion. We are conducting such scenarios now. The result will be discussed in a future paper since the focus of current paper is presenting the field survey result and validating the numerical model package.

**Comment 6**. Lines 269-270. This may be not suitable since according to IPCC, the frequency and intensity of future typhoon is increasing. Hence, the worst-case scenario of future typhoon should be more severe than typhoon Hato.

**Author's response:** We acknowledge the possibility that more severe and intense typhoons could happen in future with the changing climate. However, the objective of the numerical experiments is mainly to show the effects of tidal level and SLR on coastal inundation and we believe using Typhoon Hato's atmospheric condition as a benchmark scenario can well serve the purpose.

**Comment 7**. In conclusion, please point out clearly that the inner harbor area is the most fragile region which could be inundated even under the lowest tidal level. For this, immediate attentions/engineering actions should be took by the local government.

**Author's response:** Thanks for the suggestion. We have repeated this key point in both the result and conclusion sections.

Takagi, H., Xiong, Y., and Furukawa, F.: Track analysis and storm surge investigation of 2017 Typhoon Hato: were the warning signals issued in Macau and Hong Kong timed appropriately?, Georisk:

Assessment and Management of Risk for Engineered Systems and Geohazards, 1-11, 10.1080/17499518.2018.1465573, 2018.

Response to Reviewer 2

We thank reviewer 2 for the valuable suggestions. In this revised version, we have made changes according to the suggestions and comments and highlighted areas where those changes are made.

The point-by-point replies to the comments are below.

**Comment 1**: line 17-23: The purpose of this paper and the conclusion are suggested to be included in the abstract.

**Author's response:** Thanks for the suggestion, we actually included the purpose and conclusion in our initial version, but the abstract for NHESS has 100 word limits, so we shortened it to current version.

**Author's change to manuscript:** no change is made.

**Comment** 2. line 39: "Macau (and Hong Kong) commonly experience about 5-6 typhoons per year. . .", references are required.

**Author's change to manuscript:** we add the reference: Lee, E. K. S., L. Fok, and H. F. Lee (2012), An Evaluation of Hong Kong's Tropical Cyclone Warning System, *Asian Geographer*, *29*(2), 131-144, doi:10.1080/10225706.2012.7

**Comment** 3. line 40: ". . .by storm surges during major typhoons.." The periods are doubled.

**Author's change to manuscript:** Extra "." deleted.

**Comment** 4. line 49: ". . .Although, Macau has 2 tidtidal levele gauge". Typo.

**Author's change to manuscript:** Corrected.

**Comment** 5. line57: "which will be discussed in this paper." Redundant.

**Author's change to manuscript:** Redundant sentence is deleted.

**Comment** 6. line 66-67: "The sudden intensification occurred because of the low vertical 67 wind shear and the high sea surface temperature of ~31∘C in the Northern portion of the SCS". References are required.

**Author's change to manuscript:** we add the reference: HKO: Super Typhoon Hato (1713), Hong Kong Observatory, http://www.weather.gov.hk/informtc/hato17/report.htm, 2017.

**Comment** 7. line 68: "It is well-known that the . . . during a typhoon landfall plays ˇ . . . ". Missing words.

**Author's change to manuscript:** we added the missing word "tide". Here is the corrected sentence: "It is well-known that the **tide** during a typhoon's landfall plays a significant role"

**Comment** 8. line 70: "OSU TPXO-atlas8". References are required.

**Author's change to manuscript:** we add the reference: Egbert, G. D., and S. Y. Erofeeva (2002), Efficient Inverse Modeling of Barotropic Ocean Tides, *Journal of Atmospheric and Oceanic Technology*, *19*(2), 183-204, doi:10.1175/1520-0426(2002)019<0183:eimobo>2.0.co;2.

**Comment** 9. line 79-80. . .: "SCHISM (Semi-implicit Cross-scale Hydroscience Integrated System Model). . ."

**Author's change to manuscript:** we add the reference: Zhang, Y. J., F. Ye, E. V. Stanev, and S. Grashorn (2016), Seamless cross-scale modeling with SCHISM, *Ocean Modelling*, *102*, 64-81.

**Comment** 10. line 80: "Weather Research and Forecasting (WRF) model. . ."References are required.

**Author's change to manuscript:** We add the reference: Skamarock, W. C., Klemp, J. B., Dudhia, J., Gill, D. O., Barker, D., Duda, M. G., Huang, X.-y., Wang, W., and Powers, J. G.: A Description of the Advanced Research WRF Version 3, National Center for Atmospheric Research, Boulder, Colorado, USA, NCAR TECHNICAL NOTE, 2008.

**Comment** 11. Fig 2: Table S: The location of the photo shall be added in the figure.

**Author's response:** We added the longitude and latitude of each photo in the figure.

**Author's change to manuscript:** Please refer to Fig 2 for the changes.

**Comment** 12. Fig 3a: The red texts are not clear.

**Author's change to manuscript:** We changed the colour to yellow which we believe is clearer.

**Comment** 13. line 106-107: The locations, Ponte Pou Heng on Avenida de Demétrio Cinatti, Rua Visconde Paco de Arcos, Rua do Almirante Sergio, are missing on the figures.

**Author's response:** We realize it might be very difficult for readers who are not familiar with the roads' names to identify them on the figure. Therefore, instead of adding the names on the map, we chose to remove the names and use other features already on the maps to describe the area referred to.

**Author's change to manuscript:** The sentence with the listed names is change to "…was completely flooded to a depth of 3.1 m at Ponte Pou Heng (the purple dot). Along the coastal roads of the Inner Harbor, inundation depth reached 2.0 - 2.5m in many low-lying places."

**Comment** 14. Fig 3b: No 'Rua Do Camboa' on the figure.

**Author's response:** The street "Rua Do Camboa" is shown in Fig 3a and Fig 3d.

**Comment** 15. line 133-143: The information of time step, domain, vertical resolution used in WRF shall be explained.

**Author's change to manuscript:** We added more explanation in the manuscript: **"**There were 45 vertical levels with the lowest level approximately 50 m above the surface. The output time interval of wind and pressure fields is 10 minutes."

**Comment** 16. line 152-154: The way to determine the Manning coefficient shall be explained with related references.

**Author's change to manuscript:** We added the requested information in the manuscript: "The values of Manning coefficient are informed by previous studies (e.g. Martyr et al., 2013; Garzon and Ferreira, 2016). We choose relatively low Manning value for the estuary and open sea area as the sediment in the Pearl River Estuary is dominated by very fine sand (mainly silt clays) (Jiang et al., 2014)."

17. line 160-161: How to setup the radiation stresses in SCHISM?

**Author's change to manuscript:** The WWMIII is dynamically coupled with SCHISM every 600 seconds. The radiation stress is estimated according to Roland (2008) based on the directional spectra itself. The radiation stresses computed in WWMIII are transferred to SCHISM at each step to update water level and velocity, which are returned to WWWIII as feedback.

18. Fig 5a: The text on the figure are not clear enough.

**Author's change to manuscript:** We have modified the figure by adjusting the colour of coastline and station name. The texts now stand out from the background colours.

19. Figure 6b : What are the reasons that the error at Chiwan is larger than other stations?

**Author's response:** The tidal gauge in Chiwan is located deep inside of the bay area (see the figure below) where the bathymetry and coastal geometry were subjected to significant changes in the past decade. The bathymetric data used in the simulation does not necessarily capture the most updated status in the bay.

[Figure]

Figure The location of Chiwan gauge

20. line 195, Fig 6c, Fig 6d: The location name cannot be found on the figures.

**Author's response:** The two locations "Inner Harbor" and "Fai Chi Kei" are marked on Fig 3b. Readers can refer to Fig 3b for the locations.

**Author's change to manuscript:** We add the sentence "(see the locations in Figure 3b)" in the main text for readers' convenience.

References:

[revised manuscript text omitted]

Longitude: 113.540481
Latitude: 22.202719

[Figure]

Longitude: 113.53676
Latitude: 22.199813

[Figure]

Longitude: 113.536548
Latitude: 22.198702

[Figure]

Longitude: 113.536638
Latitude: 22.19873

[Figure]

Longitude: 113.536192
Latitude: 22.194345

[Figure]

Longitude: 113.5339
Latitude: 22.18999

[Figure]

Longitude: 113.5368
Latitude: 22.196137

[Figure]

Longitude: 113.538539
Latitude: 22.194066

**Figure 2.** Photos taken during the field survey on the Macau Peninsula.

[Figure]

**Figure 3.** (a) Measured inundation depths on the Macau Peninsula shown on a Google Earth image. (b) High-resolution bare ground elevation with marked locations. (c) and (d) Profiles of surveyed inundation water depths along two main roads: Avenida de Almeida Ribeiro and Rua do Gamboa.

[Figure]

**Figure 4.** (a) The numerical simulation domain for SCHISM-WWMIII with close-ups showing (b) the mesh in PRD

and (c) the mesh near Macau.

[Figure]

    **Figure 5.** Wind fields generated by the WRF model: (a) In the Pearl River Estuary at 12:50 PM on August 23; (b)

The wind gauge locations of PG, PN and PV in Macau; (c) – (k) Comparisons of numerical results (WRF) with measured wind speed at different locations. Locations and names of wind gauges (c) - (h) are shown in Figure 5a.

**Figure 6.** Numerical results capture well the key features of storm flooding induced by Typhoon Hato. (a) The simulated maximum surge height in the PRE; (b) A comparison of simulated and measured storm tide at four selected tide locations (marked with the green dots in Figure 5a). Note the measured data at Zhuhai station is not complete due to a power cut; (c) The surveyed inundation depths (the colored dots) overlaid on the simulated maximum inundation depths in the Macau Peninsula; (d) The arrival time of the flood wavefront.

[Figure]

**Figure 7.** Maximum inundation depths for (a) the benchmark scenario; (b) the highest extreme tide and (c) the lowest extreme tide under the current sea level.

[Figure]

**Figure 8.** A map showing the difference between the maximum inundation during the benchmark scenario (Figure 5a) and the highest extreme tide under the current sea level.

[Figure]

**Figure 9.** Maximum inundation depths during the highest extreme tide under (a) 0.5-m SLR and (b) 1-m SLR.

Maximum inundation depth during the lowest extreme tide under (c) 0.5-m SLR and (d) 1-m SLR; A map showing the difference between the maximum inundation during the benchmark scenario (Figure 7a) and the highest extreme tide under (e) 0.5-m SLR and (f) 1-m SLR.

[Figure]

**Figure 10.** The difference in maximum inundation depths between scenarios with and without the wave model during the highest extreme tide under (a) the current sea level (b) 0.5-m SLR, and (c) 1-m SLR.